# Relevance of HBx for Hepatitis B Virus-Associated Pathogenesis

**DOI:** 10.3390/ijms24054964

**Published:** 2023-03-04

**Authors:** Anja Schollmeier, Mirco Glitscher, Eberhard Hildt

**Affiliations:** Department of Virology, Paul-Ehrlich-Institut, 63225 Langen, Germany

**Keywords:** HBV, HBx, HCC, chronic infection, host factors, signaling, pathogenesis

## Abstract

The hepatitis B virus (HBV) counts as a major global health problem, as it presents a significant causative factor for liver-related morbidity and mortality. The development of hepatocellular carcinomas (HCC) as a characteristic of a persistent, chronic infection could be caused, among others, by the pleiotropic function of the viral regulatory protein HBx. The latter is known to modulate an onset of cellular and viral signaling processes with emerging influence in liver pathogenesis. However, the flexible and multifunctional nature of HBx impedes the fundamental understanding of related mechanisms and the development of associated diseases, and has even led to partial controversial results in the past. Based on the cellular distribution of HBx—nuclear-, cytoplasmic- or mitochondria-associated—this review encompasses the current knowledge and previous investigations of HBx in context of cellular signaling pathways and HBV-associated pathogenesis. In addition, particular focus is set on the clinical relevance and potential novel therapeutic applications in the context of HBx.

## 1. Introduction

### 1.1. Relevance of HBV for Global Health

Among the five different hepatitis viruses, A–E, the hepatitis B virus (HBV) presents as the most impactful on society, even preceding the hepatitis C virus in its absolute global death toll per year. According to the latest Global Burden of Disease (GBD) report in 2019 [1], HBV-induced pathogenesis still ranks 22nd place out of 177 different Level 3 causes of death, which are classified as specific morbidities such as different viral infections, diseases or third party-induced deaths. Based on the GBD report, this translates to over 320 million cases associated with over 0.55 million fatalities per year. While the overall number of cases has been reduced by a fourth in the last three decades, no significant improvement has been achieved with respect to HBV’s ranking in Level 3 causes of death as compared to 1990′s GBD report. Globally seen, the burden of hepatitis B is greatest in Sub-Saharan Africa and Southeast Asia, yet lowest in high-income countries according to 2019′s GBD report. Generally, male individuals suffer from a higher increase in years of life lost (YLL) as compared to females, which peaks in the age group of individuals between 50 and 54 years. In numbers, about 18.2 million individuals are affected by hepatitis B virus-related health complications, which are subdivided into the contributing morbidities of acute hepatitis (~8.9%), liver cirrhosis (~59.2%) and liver cancer (~31.9%) [1]. While acute hepatitis, worryingly, is the leading cause for an increase in YLL in post-neonatal infants, the most severe impact of HBV on the health of the general public is mediated via cirrhosis, liver disease and liver cancer. Obviously, the progression from an acute infection to a hepatocellular carcinoma (HCC) relies on heavy changes within the equilibrium of cellular homeostasis in infected tissues. In the past years, more and more evidence has accumulated that the viral protein HBx fulfils an essential role in shifting cellular equilibrium towards pathogenesis, which we aim to summarize and highlight in this review.

### 1.2. Viral Characteristics of HBV

HBV is a viral species classified as an enveloped virus with a partially double-stranded, circular DNA genome within the family of Hepadnaviridae and is characterized as a strictly species- and tissue-specific virus. The human pathogenic genotypes A to J (gtA-J) are clustered in the genus *Orthohepadnavirus*, which also contains viral species infecting other mammals such as bats, ground squirrels or shrews [2]. All members of *Orthohepadnavirus* comprise a genome, being ~3.2 kilobases (kb) in size, containing four overlapping open reading frames (ORFs). Upon entry into hepatocytes, which is mediated via Na+-taurocholate co-transporting polypeptide (NTCP) and other co-factors summarized elsewhere [3], the viral relaxed circular DNA (rcDNA) enters the nucleus. Here, it is converted into the episomal circular covalently closed DNA (cccDNA) via DNA-repair mechanisms [4], which serves as a template for transcription into pre-genomic RNA (pgRNA), the precursor of viral rcDNA and four subgenomic viral RNAs [5]. Notably, the nuclear amount of cccDNA can be increased through transport of de novo-synthesized genomes in an infected cell, which also elevates the efficiency of integration of viral DNA into the host genome [6]. From the five viral mRNAs, seven viral proteins are translated: (i) HBVcore, capsid protein (HBcAg), (ii) pre-core, secreted variant of HBcAg (HBeAg), (iii) viral polymerase (P), (iv) large surface antigen/protein (L-HBsAg), (v) middle surface antigen/protein (M-HBsAg), (vi) small surface antigen/protein (S-HBsAg) and (vii) regulatory X-protein (HBxAg). While HBeAg can be translated N-terminally to form the pre-core mRNA, S-HBsAg can be N-terminally extended via the preS2 or preS1/S2 domains to form M-HBsAg or L-HBsAg from another mRNA, respectively. Together with the viral pgRNA and P, HBcAg forms icosahedral capsids, which are then enveloped via the three forms of HBsAg at the endoplasmic reticulum (ER) or at multivesicular bodies (MVBs). Virions are finally released as ~42-nm-sized particles via MVBs in an endosomal complexes required for transport (ESCRT)-dependent manner [7]. Within the capsid structure, P then progresses to reverse transcribe the pgRNA into the genomic DNA, thereby forming the mature, so-called Dane particle first described in 1970 [8]. Notably, the reverse transcription is independent from capsid envelopment and can also occur in intracellular capsid structures shuttled towards the nucleus. Besides classical viral particles, HBV can even be released via exosomes [9], whereas HBsAg can also be secreted as subviral particles (SVP), forming both spheres and filaments with varying contents of L-, M- or S-HBsAg [10]. Here, filaments make use of MVB-mediated release, while spheres use the secretory pathway.

### 1.3. Pathogenesis

HBV is a hepatotropic virus, using liver tissue as the primary site of infection. According to the World Health Organization (WHO), about one-third of infected patients display an asymptomatic infection, while another third develop a non-icteric hepatitis. The remaining third display classical signs of an icteric hepatitis, including fatigue, nausea, hepatomegaly and jaundice. About one percent of these individuals may develop a fulminant hepatitis, causing heavy liver damage and death [11]. Apart from the acute course of infection, HBV infections carry a high risk of chronification, especially if infection occurs at a very young age. In this case, liver pathogenesis conventionally follows the route from inflammation over fibrosis to cirrhosis and, finally, the formation of an HCC.

Chronic infections display a different course on a scale of up to years after infection as compared to shorter, acute infections. Chronicity is mainly characterized by the detectability of viral DNA and HBcAg in serum [12]. Here, liver fibrosis is limited, while HBV DNA, HBsAg and HBeAg levels remain high over time in the immune-tolerant phase. Subsequently, the immune-clearance phase is characterized by significant signs of liver damage, thus giving rise to an increase in serum alanine transaminase (ALT) levels. Notably, low levels of HBsAg in HBeAg-positive patients indicate a higher level of fibrosis [13], thereby increasing pathogenesis. At the end of the immune clearance, seroconversion from HBeAg positivity to HBeAg negativity occurs, which is due to mutations being acquired in the viral basal core promoter (BCP) or pre-core (PC), controlling HBeAg expression [14]. This is accompanied by a reduction in HBsAg, HBV DNA, the amount of intrahepatic cccDNA and the build-up of anti-HBeAg IgG. On the side of HBsAg, however, a dependency on the HBV genotype can be observed, with gtA and gtE displaying the highest HBsAg amounts as compared to gtC, gtD and especially gtB [15]. Symptoms fade over time, leading to the stage of an inactive carrier. In 20–30% of chronically infected patients, the infection may eventually be re-activated, leading to an increase in liver damage as well as detectability of both HBV DNA and HBsAg in sera. Most frequently, re-activation of HBV is mediated via chemotherapy, immunosuppression or defects in the immunological control of the infection by the adaptive immune system [16]. This switch from inactive carrier to re-activation may fluctuate over time, thereby promoting liver damage and progression of disease [17,18]. This can also result in a so-called occult infection, which is characterized by an absence of HBsAg and viral DNA, yet the presence of anti-HBsAg IgG [19]. A special case is represented by gtG, which, per se, does not release HBsAg in vitro [20]. Pathogenesis, in general, differs in severity with respect to environmental and other risk factors, which shall be summarized hereafter.

## 2. Hallmarks of HBV-Induced Chronic Liver Disease Pathogenesis

### 2.1. General Factors Associated with Disease Progression

Several risk factors fueling the course of pathogenesis have been identified thus far [1]. This is either due to an additional, generalized risk of developing cancer or due to an elevated risk of contracting HBV. Further, the rate of patients developing a chronic HBV infection is highest upon vertical transmission as compared to horizontal transmission routes. A strong bias of males being at risk of developing an HCC upon chronic infection and liver cirrhosis exists, which may be partly due to the expression of sex-hormone receptors promoting HBV replication [21,22].

Apart from environmental factors, genetic predisposition plays a key role in the progression of a chronic HBV infection. These mainly present in the form of different single nucleotide polymorphisms (SNPs), haplotypes or differential expression in host factors required for HBV replication, such as NTCP [23], and in genes controlling antigen presentation (e.g., human leukocyte antigen, HLA), cytokines (e.g., interleukins, ILs, and interferons, IFNs), microRNAs (mi RNAs) or general modulators of innate and adaptive immunity, which are reviewed elsewhere [24]. These also come into play in the classical route of a failure in clearance of HBV, which is due to a T-cell exhaustion on cluster of differentiation 8-positive (CD8+) T-lymphocytes mediated by prolonged presence of viral antigens, causing a dysfunctionality of these lymphocytes [25].

This is also true for co-infections with human immunodeficiency virus (HIV), as a defective viral clearance increases liver inflammation and disease progression, which then also leads to impaired cluster of differentiation 4-positive (CD4+) T-cell-mediated viral clearance [26,27,28]. When looking at other viral co-infections, a similar picture arises. Hepatitis C virus (HCV), for instance, also causes T-cell exhaustion by itself, which is more pronounced in the case of a co-infection with HBV. A restoration of functional T-cells within these patient groups is currently the topic of plentiful studies, which aim to increase cytokine production, pathogen recognition receptor (PRR)-mediated signaling, antigen presentation or underlying metabolic pathways [29]. Additionally, HCV co-infections drastically accelerate pathogenesis towards a chronic liver disease and can even increase the risk of HBV re-activation [30]. While this was not frequently observed for co-infections with hepatitis A virus (HAV) and hepatitis E virus (HEV), they still pose a risk to chronic HBV patients due to augmented pathogeneses [31,32]. Among the viral co-infections in the context of chronic hepatitis B, a chronic infection with hepatitis D virus (HDV), a satellite virus to HBV, undoubtedly presents as the most severe form described thus far. Here, all routes of pathogenesis are highly enhanced, leading to severe liver inflammation and fast progression towards HCC [33].

When looking at viral factors, the ten different HBV genotypes display differing grades of disease progression. Here, gtC and gtD are described to have the highest rate of promoting HCC, which goes along with a lower rate of HBeAg seroconversion [34]. This may partly be due to a more efficient viral clearance via the immune system in the case of gtA, although gtD also induces a robust inflammatory response [35]. In addition, co-infections with other HBV genotypes could also occur naturally and results often in advanced fibrosis and cirrhosis. In particular, gtG requires a co-infection with other genotypes, most commonly gtA, for efficient virus replication [36].

With respect to viral nucleic acids, a high genomic copy number in the serum of patients indicates a more severe progression of disease [37]. The same is true for the amount of cccDNA within hepatic tissue, although a liver biopsy is more laborious and harmful for the patient as compared to serum markers [38]. Interestingly, recent studies indicate that serum HBV pgRNA is an applicable marker for severity of disease, even under antiviral treatment, as it reflects intracellular cccDNA pools [39,40]. Interestingly, inflammatory processes within the immune-clearance phase of pathogenesis are majorly linked to viral transcription from the cccDNA and not from integrated HBV DNA [41].

Apart from these variations, genomic integration of HBV significantly contributes to genomic instability, thus favoring progression towards HCC. Integrates are mainly sources of HBsAg and HBx, yet are incapable of producing full viral genomes as no circulation is possible [42]. A distinctive pattern for integration sites within the host genome does not seem to exist, yet somewhat preferred insertion hubs at, e.g., the genes telomerase reverse transcriptase (*TERT*) and lysine methyltransferase 2B (*KMT2B*) indicate a certain role of integration for tumorigenesis [43,44]. It is, however, noteworthy that previous assumptions on genomic integration, e.g., integration at cytosine-phosphate-guanosine (CpG) islands as a cause for oncogenesis by itself, need to be re-evaluated [45]. Specifically, a recent study suggests that integration at very specific sites, thereby altering expression of, e.g., tumor suppressors, is less likely to be the cause of oncogenesis, yet, rather, a consequence of a prolonged chronicity [46].

With respect to viral proteins, HBeAg is one of the most important markers for disease severity, as HBeAg-positive phases of infection are characterized by an increase in liver damage as compared to HBeAg-negative phases [47,48]. It is therefore often used as a prognostic marker, with analyses being based on acquired BCP or PC mutations. Apart from these, a recent study identified mutations in the Kozak sequence preceding PC as further prognostic markers yielding low levels of HBeAg [49]. Interestingly, these acquired mutations differ between HBV genotypes and are not strictly linked to a long phase of being an HBV carrier, as even children may carry them [50]. Mechanistically, HBeAg is associated with the expression of inflammatory cytokines [51] and is generally seen as an immunomodulatory protein [52]. Recent studies also imply that HBeAg is involved in activating local immune cells, resulting in the activation of CD8+ T-cells, and may therefore contribute to T-cell exhaustion [53].

Just as HBeAg, the level of HBsAg in chronic HBV may be used as a potential diagnostic and prognostic marker, as low levels of the viral antigen correlate with a higher level of fibrosis, yet not cirrhosis in HBeAg-positive carriers [54]. Apart from this, high levels of HBsAg offer a certain degree of immune evasion from B-cell responses, as SVPs efficiently capture anti-HBsAg antibodies, which would otherwise lead to a neutralization of HBV [55]. On the other hand, higher levels of intracellular HBsAg may lead to a more pronounced tumorigenesis via oncogenic signaling cascades such as Ras-GTPase-, Phosphoinositide 3-kinases (PI3K), cyclin-dependent kinase (CDK)- or ER-stress pathways [56,57,58,59,60]. While HBsAg represents one of the major host modulatory viral proteins, HBx fulfils even more regulatory roles with respect to pathogenesis and disease progression.

### 2.2. Clinical Significance of HBx

While HBx was not considered relevant for the life cycle of HBV in the past [61], it soon became apparent that it does display transforming properties, thus being able to promote tumorigenesis [62]. The first reason lies within its capability of boosting HBV replication, e.g., on the basis of HBx-dependent cccDNA formation [63] or the modification of cccDNA-associated histones [64,65]. Further, HBx has been described to be inhibited by estradiol benzoate [66], which may, in part, contribute to the trend of higher rates of disease progression of chronic HBV in male patients.

The role of HBx in cccDNA control may be made use of in histological analyses, where nuclear HBx, suggesting an active role in controlling cccDNA pools and in contrast to other viral proteins, seems to be well expressed and is associated with the grade of recurrence [67]. This is in line with previous findings pointing out a direct correlation between HBx expression and the likelihood of developing cirrhosis preceding HCC [68]. The reason for this may lie within an increase in integrates of viral DNA into the host genome. As these are sources of HBx, it is not surprising that there is a certain build-up of HBx, yet also anti-HBx IgG in HCC and chronic HBV patients over time [69]. Interestingly, this can be observed rather frequently and at early stages of infection [70]. This goes along with the HBeAg seroconversion, as distinct mutations in HBx (positions 127, 130, 131 and 132) correlate with a reduced amount of HBeAg, thus also with disease progression [71,72,73]. Apart from loss of HBeAg being linked to HBx detectability, a correlation between the amount of HBsAg and HBx can be found, with loss of the first causing a rise in the latter [74]. Interestingly, mutations or truncations within the C-terminal region were identified to induce cell growth as well as build-up of reactive oxygen species (ROS), which may be the reason for the enhanced pathogenesis [75]. Here, HBx may act positively on growth-factor signaling mediated via human epidermal growth factor receptor 2 (HER2) [76] and on telomerase activity by inducing the expression of TERT [77]. On the other hand, these truncations were also found to be associated with beneficial outcomes in immune prophylaxis of perinatally infected infants due to a decrease in viral replication [78], which points out that further studies are needed to disambiguate the situation.

In summary, more and more evidence accumulates for HBx being of central interest when it comes to evaluation of the severity of pathogenesis. However, a more detailed understanding of underlying mechanisms is urgently required. How significantly the C-terminal region and other parts of the protein contribute to chronic liver disease and HCC formation on the basis of altering signaling pathways shall be addressed in this review.

## 3. Properties and Challenges of the HBx Protein in Context of HBV-Associated HCC

HBx is a multifunctional protein with a pleiotropic activity in numerous viral and cellular pathways. The protein is 154 amino acids in size and gained its named due to the lack of sequence homology with other known proteins. However, HBx shares a conserved homology (about 72%) with all mammalian Hepadnaviruses, although, curiously, being absent in Avihepadnaviruses [79]. First, in 1987, the HBx gene was suspected to encode a transactivating factor [80], which is conserved over the integrated form of HBV DNA [81]. Nevertheless, as of yet, the flexible nature of HBx combined with difficulties in a suitable and robust experimental system [82] make it challenging to study the function and fundamental biological relevance of HBx in the development of hepatocellular carcinogenesis.

### 3.1. Characteristics of the HBx Protein

The primary structure of the x-protein is subdivided into two distinct functional domains. The amino terminal (N-terminal) acts as negative trans-activator domain (aa 1–50) and includes a proline-/serine-rich area (aa 30–46). The carboxy terminal (C-terminal) domain acts as transcriptional activator or co-activator domain (aa 52–148) and comprises mitochondria localization sites (aa 54–70; 75–88; 109–131) as well as numerous target regions for interaction with host proteins [83]. Moreover, it has been identified that the trans-activator domain between amino acid 58 and 119 is essential for signal transduction, whereas the subsequent part, between amino acid 120 and 140, is crucial for nuclear trans-activation [84,85].

While the organization of HBx into certain regions fulfilling different functions, along with their putative effects on viral and host processes promoting hepatic injury, are well known, the secondary structure or even a three-dimensional model is yet to be identified. Nonetheless, the N-terminus was recently characterized as a highly conserved region displaying characteristics of an intrinsically unfolded and unstructured protein [86]. On the other hand, the C-terminus harbors a more organized region with key structural regions required for most of the functional activity of the protein: a zinc-binding motif formed by a highly conserved cysteine and histidine region [87], a conserved alpha-helix H-box motif [88] and a BH3-like domain [89]. Several studies also reported the formation of HBx homodimers via disulfide bonds and acetylation, although the detailed dimerization and oligomerization process is controversially discussed [90,91]. Apart from structural features, the molecular characterization of HBx revealed a relatively low expression level in infected livers and, moreover, only a minor amount of soluble protein in a recombinant system [92]. Both characteristics lead to the failure of the attempt to crystallize full-length HBx, thus leading to the lack of a three-dimensional structure [93]. In a different approach, Van Hemert et al. published in 2011 an in silico-generated model of the tertiary structure of HBx, which was found to share similarities with the central domain of DNA glycosylase [94]. In addition, Zhang et al. were able to solve the crystal structure of the HBx–BH-3-like motive in complex with BCl-xL. Hereby, three residues of the HBx–BH3-like motive were observed to be located on one side of a short α-helix and fit into a hydrophobic pocket of the Bcl-xL protein, which is a relevant aspect for the HBV life cycle and cytotoxicity in hepatic cells. This study provided the first important insights into the much-needed structural knowledge of HBx and promoted understanding of HBx-induced liver disease [95].

Besides the full-length or canonical HBx, the viral ORF encoding HBx also comprises two in-frame translation initiation codons to generate multiple N-terminally divergent isoforms of the protein [96]. These initiation codons of HBx mRNA seem to be highly conserved over the different HBV genotypes [97]. Moreover, the individual HBx isoforms are associated with different and overlapping subcellular localizations within the cell. In general, the location of HBx is based on its relative abundance. At low expression levels, the canonical HBx is preferentially nuclearly located, yet shows predominantly cytoplasmic accumulation at higher expression levels. However, the N-terminally truncated, middle isoform behaves opposingly to this, with low expression causing a cytoplasmic localization, yet high expression inducing a nuclear distribution. Lastly, the short HBx isoform retains mainly cytoplasmic distribution, independent from the expression level. Recently, Prieto et al. reported an additional regulatory mechanism that governs nucleocytoplasmic distribution of HBx, which is mediated through the phosphorylation of phylogenetically conserved residues within the protein [98].

In light of this observation, the dual localization and lack of the negative regulator domain (middle isoform) and parts of the trans-activator domain (small isoform) in different pools of HBx once more explain the multifunctionality of this viral protein [99,100]. These give rise to the differential localization, where nuclear HBx mainly regulates transcription of viral and host genes, whereas cytoplasmic HBx mainly targets signaling cascades and mitochondria. Through this, HBx majorly controls hepatocellular pathogenesis and development of HCC [101,102,103].

### 3.2. Technical Limitations and Challenges in HBx-Associated HCC Research

The strict hepatotropic property of HBV and the long-standing limitations in infection models together with the unique characteristics of HBx itself provide major challenges in gaining physiologically relevant insights into HBx function during HBV replication and associated oncogenic effects. Consequently, many aspects of HBx are still unknown and causal for controversial reports about its relevance. One problem is that the human HCC cell lines HepG2 and Huh7 are often used as platforms for in vitro studies. While both cell lines are permissive for HBV replication and virion production, they are not suitable for HBV infection [104].

To address questions surrounding the natural course of infection, either primary hepatocytes (derived from human PHH or the tree shrew, *Tupaia belangeri,* PTH) or the differentiated HepaRG cell line is used. These, in turn, are limited by several drawbacks such as low reproducibility and permissiveness to HBV infection [105,106]. The identification of hNTCP as a functional entry receptor for HBV infection in 2012 overcomes this limitation and provides a powerful tool for in vitro studies of HBV infection, and thus for studying HBx-dependent effects on cellular pathways [107,108]. The overexpression of hNTCP in Huh7 and HepG2 provides susceptibility for HBV infections [109,110,111]. In addition, a recent study used the hepatocyte cell line BEL7404. These cells can be transfected with a high transfection efficiency and display a higher viral antigen expression after HBV-replicon transfection compared to the common human liver cell lines. While overexpressing hNTCP along with the inducible hepatocyte nuclear factor 4α (HNF4α), BEL7404 cells demonstrate efficient HBV replication and susceptibility for an HBV infection [104]. Additionally, novel approaches comprise an optimization of previously established methods [112,113,114] or the generation of novel hepatocyte-derived culture models. These include the generation of human induced pluripotent stem (iPS) cell-derived hepatocyte-like cells (iPS-HLCs) or the development of liver organoids as a 3D in vitro model for HBV-associated liver cancer studies [115,116,117,118,119]. In vivo animal studies are restricted due to the high species tropism that can be naturally infected by HBV, which encompasses humanized mouse models, macaques, tree shrews or other mammals, as reviewed elsewhere [120,121,122]. Commonly used is also the woodchuck hepatitis virus (WHV) model, which shares similar infection kinetics to the human HBV infection course [123].

Mainly, HBx-specific investigations are based on in vitro studies of transient transfections with HBx-encoding plasmids. The overexpression in HepG2 or Huh7 cells provides the opportunity to investigate the direct function and cellular protein interactions of HBx, yet is limited by representative physiological conditions. Hereby, the absence of other viral proteins and artificial subcellular localization of HBx, depending on the expression level, offers a critical aspect for the investigation of distinct biological pathway analyses [82].

These obstacles can be partially overcome by a transient transfection of plasmids harboring an overgenomic HBV construct (e.g., 1.5-mer) [124]. This helps to gain a better understanding of HBx in the context of viral replication and virion assembly, but it is tremendously affected by cell line-specific effects of HBx. Another option frequently used is epitope or fluorescently tagged HBx constructs for the transient transfection systems [82], where tags are necessary due to restrictions in the availability of commercial HBx antibodies. However, the impact of tags and relatively large fluorescent proteins on the conformation and function of the only 17 kDa in size protein is speculated. Recent studies therefore designed an HBx reporter construct in combination with the binary luciferase reporter assay (NanoLuc-HiBiT detection system) to achieve minimized, tag-specific obstacles without affecting HBx function, but allowing a robust high throughput screening of novel HBx-specific targets [66,125].

A different approach is considered in some studies, where HBx was produced in *E. coli* as a recombinant protein with a His-tag and the preS-derived translocation motif (TLM), which yields cell-permeable HBx [87,126] and avoids troubles of interfering plasmid DNA from an HBx transfection system.

To gain more valuable information at physiological levels of HBx, infection models provide a useful technique to investigate HBx-dependent impacts in the context of the entire HBV life cycle and at biologically relevant expression levels. Nevertheless, the production of integrated viral DNA fragments into the host genome, which significantly increase HBx levels during natural, persistent infection, still remains a challenge. Likewise, the investigation of HBx-dependent function during tumor progression and the influence on hepatocyte transformation is severely limited.

## 4. HBx-Induced Signaling Pathways and Phenotypic Changes as a Driving Force of Virus-Induced Pathogenesis

Signaling pathways are the key factor in the development of HCCs. HBx is known to interact with numerous cellular and viral proteins and is therefore one of the driving forces in progression of aberrant pathway signaling and hepatocarcinogenesis. The dynamic subcellular localization, based on the expression level of HBx, is the reason for the diverse function and impact on several host pathways and cancerogenic mechanisms. These include the dysregulation of processes such as metabolism, epigenetic modifications, calcium signaling, oncogenic pathways, transcriptional activation and immune response [127]. This is highlighted by a proteomic screening in HepG2 cells, which identified a total of 402 proteins interacting with the full-length form of HBx, whereas 189 proteins demonstrate a specific binding to the trans-activator domain of HBx [85], but are yet to be validated. However, many HBx-related cell-signaling pathways have only been partially elucidated, yet these provide possible targets against viral replication and HCC development and may help in the effort to gain a functional cure for HBV. In the following, main HBx-affected signaling pathways and recent investigations are summarized and clustered by their subcellular localization within the cell.

### 4.1. Nuclear Localized HBx in Context of Cancer Promoting Signaling Pathways

The subcellular localization of HBx strongly depends on its expression level. A low level of the viral protein, occurring during early phases of HBV infection, drives HBx localization within the nucleus of hepatocytes [100]. By this localization, HBx modulates the gene expression of several viral and cellular genes through activation of epigenetic molecules or interaction with chromatin-modifying enzymes as well as the basal transcription machinery [128,129]. In a genome-wide screening on mapping the binding sites of HBx in an HBV-replicating system, several protein-coding genes and non-coding RNAs (ncRNA) were identified to be directly targeted by HBx. Among cellular targets, HBx was also observed to target genes and ncRNAs boosting the viral replication [130].

The epigenetic reprograming mediated by HBx in the context of host genes as well as viral cccDNA modifications includes DNA methylation/demethylation, histone modifications, chromatin remodeling and the expression of micro RNAs, which alter cellular functions of infected hepatocytes and the related pathogenesis [131]. In addition, the epigenetic landscape provides an important factor for monitoring malignant transformation and HCC development. Coherently, these HBV-related epigenetic signatures are of fundamental interest as biomarkers for therapeutic applications [132,133].

In line with this, HBx-mediated abnormal DNA methylation is frequently associated with the formation of HCC. On the one hand, DNA methylations encompass the genome-wide hypomethylation and activity of protooncogenes. On the other hand, regional hypermethylations on specific promotor regions and CpG island regions of tumor suppressor genes add up to these deregulations. HBx expression has been demonstrated to mediate DNA methylation mainly by the activation and deregulation of the DNA methyltransferases (DNMT) 1 and 3A [134,135]. Zhu et al. demonstrated an HBx-induced activity of the DNMT in HCC liver tissues and a direct correlation between high HBx expression levels and elevated hypermethylation in the promoter region of p16^INK4A^ in non-tissue samples. The tumor suppressor gene p16^INK4A^ is therefore a major player in tumor progression, especially in early phases of HCC [136]. In addition, HBx drives DNMT1 and DMNT3A towards the promoter region of ankyrin-repeat-containing, SH3-domain-containing, and proline-rich-region-containing protein family 2 (ASPP2) and leads to their downregulation with further effects for the p53-dependent apoptosis and HCC progression [137]. Other tumor suppressor genes, which are induced due to HBx-related hypermethylations, comprise, for example, E-cadherin (CDH1), the Ras-Association Domain Family (RASSF1A), the insulin-like growth factor-3 (IGFBP-3) and interleukin-4 receptor, which is associated with the immune response [138,139]. This is also backed up by clinical data retrieved from a comparison of HBx-positive and HBx-negative HCC patient-derived liver tissues. Here, a higher frequency of hypermethylations and methylated CpG-patterns in the IGFBP-3 promoter regions was elicited in response to HBX positivity [135]. Further, a recent study described a direct correlation of changes in the content of epigenetic methylation at the fifth position of cytosines (5mC) and their oxidized derivates with different stages of HCC tissues from HBV-infected patients [140]. Furthermore, in vivo and in vitro experiments also provide evidence for an HBx-mediated increase in DNMT activity with subsequent DNA methylation on the viral cccDNA. These data were correlated with an enhanced viral replication and nicely display the effects of HBx-driven DNA methylation on HBV itself [141]. In addition, recent studies identified a mechanism in which the HBx protein recruits the N6-methyladenosine (m6A) methyltransferase complex onto the viral cccDNA and causes co-transcriptional m6A modification of the viral RNA and also on cellular RNA loci. Especially in the case of viral RNA, m6A modifications affect the stability as well as subcellular localization of the viral transcripts and are under discussion to promote chronic hepatitis. Another m6A modification site was identified in the coding region of HBx mRNA with a negative regulatory effect on HBx protein formation [142,143].

Apart from direct DNA modification, histone modifications play a similarly crucial role in controlling gene expression, thus indicating a severe impact on aberrant regulation often observable in HCC tissues. In this regard, HBx induces the methylation, acetylation as well as deacetylation of lysine or arginine residues of histones, which heavily contributes to the effect of cancer-related genes [131,144,145]. One example is the HBx-related interaction with the histone acetyl transferase CREB-binding protein (CBP)/p300 with inducing effects on IL-8 cytokines and proliferating cell nuclear antigen (PCNA) expression [146]. Furthermore, deacetylation via histone deacetylase 1 (HDAC1) mediated by the HBx complex formation has an effect on the chromatin structure and transcriptional repression of IGFBP-3 [147].

On the side of histone methylations, more recent studies reported a high expression level of the WD repeat domain 5 protein (WDR5) in HBV patient-derived HCC tissues. In vitro and mouse models confirm an HBx-dependent implication of aberrant WDR5 levels within elevated genome-wide histone H3 lysine 4 trimethylation (H3K4me3) modifications and resulting activation of cancer-related genes [148]. In addition, Takeuchi et al. investigated an association of HBx with the histone methyltransferase suppressor of variegation 3–9 homolog 1 (SUV39h1). This led to elevated signaling in cell proliferation and ER stress in long-term infected patients [149]. While these present as important targets within the host, histone-modifying enzymes are also crucial for transcriptional regulation of the viral minichromosome consisting of the cccDNA and cellular histones. A recent study identified the specific region of the HBx protein, which is involved in the recruitment of histone-modifying enzymes (P300 and HDAC) towards the viral transcript [65]. These then control the gene expression from cccDNA in an HBx-dependent manner.

Non-coding RNAs (ncRNAs) represent another type of epigenetic modulator molecule and have gained growing attention in the past years. These seem to be key regulatory molecules for cellular gene expression with crucial involvement in tumorigenesis during HBV infection [150]. Dependent on their length, ncRNAs are classified into short ncRNAs including microRNAs (miRNAs) and long ncRNAs (lncRNAs), and they modulate tumorigenesis via different mechanisms [131,151]. Several studies identified numerous miRNAs targeted by HBx, which then have implications in liver-related genes, especially in several transcription factors, but they also trigger the dysregulation of cellular immune pathways [130,152,153]. miR-122, for instance, is the most relevant ncRNA in hepatocytes with fundamental regulatory functions. This is found to be down-regulated in HBV-related HCCs [154]. The HBx-specific upregulation of miR-21 regulates, among other pathways, the signal transducer and activator of transcription 3 (STAT3) activity and elevates the interleukin 6 level [155], thereby impacting inflammatory responses. Similarly, interactions of HBx with lncRNAs are correlated with epigenetic control of tumor-related cellular genes, yet also with the viral cccDNA [156]. One example is the lncRNA SET and MYND domain containing 3 (SMYD3), which is significantly induced by HBx and is related to an extracellular signal-regulated kinase (ERK) and protein kinase B/ glycogen synthase kinase 3 beta (AKT/GSK3-β) signaling activation, resulting in aberrant cell proliferation and metastasis with poor HCC prognosis [157].

In conclusion, growing evidence shows a tremendous effect of aberrant epigenetic modifications caused by nuclearly located HBx. This, therefore, is of major interest not only as an HCC-related biomarker, but might also play a key-role in conferring cancer drug resistance [158]. However, both the effect of expression level on nuclearly localized HBx as well as the unknown differences of HBx derived from different HBV genotypes keep serious gaps in the knowledge of epigenetic modifications. This requires further studies, as promising benefits for therapeutic strategies may arise from a more detailed understanding.

### 4.2. Cytosolic HBx-Mediated Signaling Pathway Regulation and Pathological Effects for the Liver

The interaction of HBx with cellular proteins is the driving force for pathological progression in hepatocytes and is strongly connected to the development of HBV-mediated HCC. HBx predominantly localizes in the cytoplasm. Here, it interacts with several transcription factors, followed by nuclear import and proteins involved in key cellular signaling pathways with eminent effects on cell survival, metabolism, inflammation, genomic instability, angiogenesis and several other mechanisms. The hijacking of host cellular proteins and deregulation through the HBx then results in provoking tumor invasion, tumor metastasis, liver malignancy and, finally, hepatocellular carcinoma [127].

One of the best characterized interactions with HBx is the binding with the DNA damage-specific DNA-binding protein 1 (DDB1). This serves as an adaptor protein for the cellular cullin 4 (CUL4) E3 ubiquitin-ligase, which then mediates ubiquitination and degradation of the structural maintenance of chromosomes protein 5/6 (Smc5/6) complex. Interestingly, Smc5/6 binds to the cccDNA and acts as a negative modulator so that a degradation of Smc5/6 facilitates viral replication [159]. In addition, Smc5/6 is known for its key role in DNA damage repair mechanisms by homologous recombination, which is strongly impaired by HBx-mediated Smc5/6 degradation [160]. In addition, several signal transduction pathways such as nuclear factor-κB (NF-κB), Wnt/β-catenin, the Janus kinase (JAK) / STAT, PI3K/AKT, Ras/Raf-mitogen-activated protein kinase (MAPK), Src-kinase, p53 and many others are modulated by cytosolic HBx [127,161]. While all of the latter promote tumorigenesis via shifting metabolism, migration and survival, the first needs to be highlighted for its manifold impacts in the context of inflammation. As a master contributor to cellular processes such as immune response, proliferation, morphogenesis and cell growth, NF-κB is one of the most prominent targets of HBx. Notably, a sustained NF-κB activation is causative for a central function in development of severe liver disease, fibrosis and HCC. As reported elsewhere, HBx-induced cellular ROS levels, cytokine expression and TNF-α activation, but also a variety of HBx-mediated signal cascades of signal transduction pathways, lead to an IκBα phosphorylation and degradation. This is then followed by an induced NF-κB response for the lack of inhibition [162]. In addition, a prolonged NF-κB activity is mediated directly by HBx due to the binding and nuclear co-localization with the IκBα subunit, but also by a reduction in the level of cytoplasmic NF-κB1 [162,163]. As NF-κB then goes on to induce, e.g., inflammatory cytokines, this presents as one major part in the HBx-promoted liver inflammation, thus being one of the key drivers in chronic liver disease.

As a consequence of HBx-mediated induction of several signaling pathways, HBx also affects proliferation and disrupts cell cycle progression by deregulation of several key points controlling cell cycle progression [164]. For example, HBx regulates the levels of several cell cycle regulatory proteins such as p16, p21, p27 and cyclin D1, A and B1, which results in the activation of CDK2. Moreover, HBx induces quiescent hepatocytes to enter the G1 phase of the cell cycle. Both aspects favor cellular proliferation pathways and have fundamental consequences for cancer progression [165].

Besides the direct interaction of HBx with signal transducers and interference with distinct pathways, an HBV infection and liver dysfunction is also connected with an abnormality of hepatocellular metabolism, including metabolites such as glucose, lipids, and amino acids [166]. In this regard, in vivo and in vitro studies demonstrated a reprogramming of the glucose metabolism by thioredoxin-interacting protein (TXNIP) suppression via a C-terminally truncated form of HBx with major consequences for hepatocarcinogenesis [167]. Furthermore, the hepatitis B X-interacting protein, HBXIP, also termed late endosomal/lysosomal adaptor, MAPK and mTOR activator 5 (LAMTOR5), was claimed to induce the upregulation of glucose transporter 1 (GLUT1) through the NF-κB pathway during hepatic cancer development [168]. Both mechanisms could then translate to an increase in intra-tumoral glucose levels favoring the general metabolic switch to glycolysis.

HBx further was correlated with an abnormal energy metabolism by elevated generation of ROS and consequent oxygen stress-related cell injury [169]. In light of this, HBx participates in the ER stress response as well as the mitochondrial respiratory chain, which causes an upregulation of the cytoplasmic ROS levels in hepatocytes. In particular, based on an HBx-expressing mouse model, Ling et al. investigated elevated oxidative stress in hepatocytes together with a higher expression level of specific inflammatory mediators such as interleukins and TNFs as well as with induction of connected pathways (NF-κB/AKT) [170]. Another study confirmed these observations and reported increased ROS levels in HBx transgenic mice with direct effects on metabolism and tumor progression [171]. In addition, under oxidative stress, HBx triggers a hepatic inflammatory response through the activation of the NLR family pyrin domain containing 3 (NLRP3) inflammasome, which results in production of IL1 [172]. In this context, advances in understanding HBx-driven effects on calcium signaling need to be mentioned, as these may similarly impact the inflammatory responses mediated by calcineurin, and, besides, this plays a critical part in virus replication and cellular pathogenesis. Especially, HBx was known to modulate Ca^2+^ efflux from the ER and mitochondria, thereby altering cytosolic calcium homeostasis and inducing HCC [173,174]. Previously, it was further shown that the C-terminal part of HBx interacts with the plasma membrane-localized store-operated calcium channel Orial1 and increases cytosolic Ca^2+^ influx from the extracellular space [175]. This may then translate to overall pathways such as protein kinase C (PKC)-signaling being regulated by the ions. Although an increased intracellular Ca^2+^ level may present as crucial for HBV replication, many insights into HBx-mediated calcium dynamics and related effects on tumorigenesis are largely unknown [173,176].

In conclusion, cytosolically localized HBx contributes to a countless number and complex network interaction of cellular pathways, which are schematically summarized in Figure 1. The aberrant and abnormally regulated pathways are contributors for several tumor-promoting traits and reflect the multifunctional nature and hepatocarcinogenic capacity of HBx. A question still remaining elusive is how significantly the stoichiometric ratio of HBx to other host factors and viral proteins influences signaling cascades and host processes. Similarly, solubility of HBx in physiological conditions is yet to be characterized, which will deepen the understanding of underlying pathways.

### 4.3. Impact of HBx-Dependent Interaction with Mitochondria and Endoplasmic Reticulum

Apart from its nuclear and cytosolic role, HBx also interacts with mitochondria and profoundly modulates mitochondrial morphology and function. An aberrant alteration in mitochondrial dynamics in general is associated with different pathological states of a cell. To this end, HBx is known to interact with several mitochondrial proteins and is therefore a deciding factor for disrupting mitochondrial physiology and modulating cellular signaling pathways driven by these. This includes ROS-mediated signaling, calcium signaling, apoptosis and the innate immune response, which is ultimately connected to a high impact on chronic liver disease [177]. A putative transmembrane region of HBx (aa 54–70) was recently mapped to be required for mitochondrial association, while the sequences between amino acid 75–88 and 109–131 were classified as mitochondrial target regions [178]. Based on a later study using WRL68 and HepG2 cell lines, a 44-amino-acid-long sequence in the carboxy terminal region of HBx (111–154) was shown to be sufficient for mitochondrial localization and targeting already. Especially the cysteine residue within that area (115) seems to have a key importance for the interaction with mitochondrial membrane proteins [179]. In addition, the subcellular targeting of HBx within mitochondria and its multiple effects on physiological functions is also well established in other common cell lines such as HepG2 cells as well as in mouse or rat hepatocytes [170,180,181].

HBx-associated mitochondria often correlate with perinuclear clustering, caused by a decreased, HBx-dependent intracellular motility of mitochondria along microtubules [182]. Manipulation of the dynamic balance of the mitochondrial network is another hallmark of virus-induced mitochondrial dysfunction and is correlated with central effects on cellular homeostasis and innate immune response. In light of this, HBx mediates and enhances function and translocation of the dynamin-related protein 1 (Drp1). This then translates to enhanced mitochondrial fission as an initial step to promote mitophagy [183]. In addition, HBx localizes to the voltage-dependent anion channel 3 (VDAC3) at the outer mitochondrial membrane (OMM). This leads to a decrease in the mitochondrial membrane potential and depolarization of mitochondria, which is then followed by release of cytochrome c into the cytosol and induction of apoptosis [184].

With respect to these mechanisms, several different HBx-dependent processes are now involved. On the one hand, HBx triggers the Pink1/Parkin-dependent induction of mitophagy. This leads to an escape from general apoptosis, which then promotes cell survival and virus persistence. HBx does so by inducing the expression of PINK1 on the OMM and recruitment of the ubiquitin E3-ligase Parkin to the dysfunctional mitochondria, which promotes selective degradation and prevention of apoptotic cell death [183,185]. On the other hand, depolarization of mitochondria leads to an excessive increase of mitochondrial reactive oxygen species (mtROS) and to a subsequent induction of hepatocyte inflammation [170]. Increased ROS levels, due to elevated mtROS production through damaged mitochondria or through HBx-induced cellular ROS levels, play an emerging role in mitochondrial dysfunction. Hereby, the respiratory chain of mitochondria becomes heavily disrupted and elicits a further increase in oxygen stress, which facilitates the mitochondrial translocation within the cytoplasm. Damaged, released mtDNA further contributes to the activation of signaling pathways such as NF-κB/p-AKT, AP-1 and STAT4 signaling or serves as a damage-associated molecular pattern (DAMP) to stimulate inflammatory mediators by activating toll-like receptor 9 (TLR9) and stimulating the interferon-regulated genes (STING) pathway [170,186,187]. In essence, this then promotes both inflammatory responses as well as the interferon response. Previous studies demonstrated a nexus of cell-free circulating mtDNA in serum samples of chronic infected patients with hepatocellular carcinomas [188,189], which indicates a certain potential of activating even circulating lymphocytes. In addition, a next-generation sequencing approach of mtDNA in HBV-related hepatocellular carcinomas revealed a crucial impact of specific mutations in the D-loop region of mtDNA for invasion and metastasis in liver tumors. Therefore, these aspects offer the possibility to use circulating mtDNA as a non-invasive biomarker to monitor HCC status in HBV-infected patients [190].

Furthermore, HBx interacts with Cyclooxygenase-3 (COX-3), which is part of the respiratory chain, at the inner mitochondrial membrane (IMM) and, thereby, mediates ROS generation, but also cell growth through COX-2 induction [181,191]. In contrast, the mitochondrial ubiquitin ligase MARCH5 attenuates ROS production and COX-2 activity, while mitochondria-associated HBx is degraded through proteasome degradation. Elevated MARCH5 expression levels are associated with a better survival rate in HCC patients, again highlighting the clinical significance of this mechanism [192]. However, the role of HBx as an inducing factor for apoptosis is controversially discussed. In the mitochondria, HBx is known to interact with p53 on the OMM, which then acts in a pro-apoptotic fashion. Similarly, HBx interaction with Bax causes elevated alteration in the membrane potential, thus contributing to cell death [193,194]. In addition, the binding of HBx to heat-shock protein 60 (HSP60) in the mitochondrial matrix is assumed to promote cell death. In contrast, the above-mentioned selective mitochondrial degradation and HBx-dependent suppression of SIRT4, a mitochondrial protein that normally promotes cell cycle arrest and apoptosis, argues against this statement [195]. Likewise, the upregulation of the pro-inflammatory mediators (IL-6, IL10, NF-α, etc.) by increasing ROS levels is most likely associated with an anti-apoptotic behavior resulting in tumor progression and virus persistence [170].

With respect to host defense mechanisms, mitochondrial HBx is described to interfere with the innate immune response. Here, two residues of HBx (Asn118 and Glu119) bind the mitochondrial antiviral signaling protein (MAVS) and suppress retinoic acid-inducible gene-I (RIG-I), which finally leads to a reduced antiviral immune response. Interestingly, a comparison between HBx derived from different HBV genotypes (gtA-D) revealed a heterogenic response to RIG-I-MAVS signaling, with a higher RIG-I-mediated interferon induction in gtB [196], which potentially is one factor explaining the lower rate of pathogenesis in this genotype.

As mentioned earlier, HBx also plays a crucial role in calcium signaling by hijacking the second messenger Ca^2+^ for virus replication and HBV-associated disease. The ER provides one of the major storage pools for Ca^2+^. This is why ER stress, mediated by elevated intracellular ROS levels or directly through interaction with ion channels, causes an increase in intracellular calcium signaling. Notably, the OMM and ER membrane are directly associated through organelle–organelle contact mediators. An HBx-mediated ER stress thus leads to the release of Ca^2+^, which is then translated further into an influx into mitochondria mainly by VDAC. An excessive Ca^2+^ load in mitochondria finally causes a release of the ions into the cytoplasm. Growing evidence suggests a tremendous impact of HBx on the influx of Ca^2+^ into mitochondria, which is associated with the activation of several calcium signaling pathways, which is assumed as a novel pharmacological application against HBV [173,174,197].

Conclusively, the multiplicity of HBx interactions with mitochondria and associated proteins in all parts of these organelles (OMM, IMM and matrix) highlight the emerging impact the viral protein has on signaling pathways and profound alteration of mitochondrial function. These are the major sources of ROS and are connected with a severe impact on physiology of hepatocytes and development of HCC. HBx-mediated dysfunction in mitochondrial calcium dynamics is connected with a far-reaching influence on viral pathogenesis, which is summarized in Figure 2. Therefore, HBx-mediated impact on mitochondrial dysfunction provides promising options for drug development to restrict viral replication and HCC progression.

## 5. HBx towards a Cure of HBV-Related HCCs

A chronic HBV infection is a major causative factor for the development of chronic liver disease (CLD) and HCC, which, in large parts, comes down to dysregulations mediated by the viral protein HBx. HBV thereby is responsible for approximately 900 000 HBV-related deaths annually worldwide [198]. The virological hallmarks of chronic HBV carriers are the persistence of the viral genome through the production of a stable, covalently closed circular DNA (cccDNA) as a transcriptional template for viral proteins and the ability to integrate viral DNA into the host genome [199]. Both features are ultimately associated with the activation of numerous cellular signaling pathways, which act on both necro-inflammatory and pro-tumorigenic mechanisms, thus representing a major challenge in the development of effective therapeutic targets [200].

### 5.1. Therapeutic Opportunities for HBV Treatment

Previous therapy approaches comprise the induction of antiviral immune response by treatment with pegylated interferon-alpha in combination with inhibition of the viral polymerase by nucleoside or nucleotide analogues. However, striving for a functional cure, which is defined by a clearance of HBsAg, often results in low rates of success or long-termed treatment [201]. Especially for factors such as the HBV genotype, the status of HBeAg positivity, the progressiveness of chronicity or a co-infection with HDV tremendously influence the therapeutic outcome [17,202,203,204].

Novel potential approaches comprise, for instance, the inhibition of the assembly of nucleocapsids as well as the blocking of the HBV entry receptor NTCP by monoclonal antibodies or small compounds [205,206,207]. Especially the latter aspect comes into focus of current research and provides benefits as primary targets for drugs to inhibit both HBV and HDV entry [208]. However, the elimination of the cccDNA pool and integrated viral DNA still remains unharmed and is responsible for missing the ultimate goal of an HBV-treatment—a complete cure of the virus [17]. In this regard, the majorly underestimated potential therapeutic target HBx becomes of great interest. As outlined before, HBx controls viral transcription from cccDNA to stimulate viral replication, yet also interferes with several cellular and immune-mediated pathways [64,209,210]. The latter is further enhanced by increasing levels of HBx production through viral integrates and therefore is suspected as the driving force in induction of fibrosis and HCC development [127,211,212]. Recent efforts in HBx-specific therapeutics therefore address both viral replication and the development of disease by several approaches.

### 5.2. siRNA Constitutes a Promising Therapy Approach against HBx

A potent antiviral approach was based on the use of HBx-specific siRNA and showed promising results both in vivo and in vitro [213,214,215]. A therapeutic HBV down-regulation by siRNA targeting all HBV transcripts in combination with peg-IFN-α results in a strong reduction of HBx levels and re-appearance of the SMC5/6 complex with associated epigenetic suppression of cccDNA [216].

### 5.3. HBx-Specific Monoclonal Antibodies and Therapeutic Vaccines for Treatment of Chronic HBV

The restorage of the SMC5/6 complex becomes of further importance through HBx-induced genomic instability and a potential promotion of cccDNA transcription—both hallmarks of a favored liver carcinogenesis. In light of this, a specific monoclonal HBx antibody targeting the interaction with the key cellular adaptor protein DDB1 was designed to prevent HBx–DDB1 interaction and retain the function of the SMC5/6 complex. The monoclonal antibody (mAb) was observed to recognize a highly conserved region across HBx of different HBV genotypes and is therefore promised to operate with broad reactivity and efficiency [217].

Another approach is being based on a monoclonal HBx-specific antibody fused to a cell-permeable carrier molecule. This allows targeting of intracellular HBx and facilitates the reduction of HBV transcription and viral proteins through TRIM21-mediated protein degradation in vivo and in vitro. This method overcomes limitations of monoclonal antibodies targeting intracellular molecules, which are due to the lack of membrane permeability of mAbs [218].

In contrast, Horng et al. studied the impact of HBx as an immunogen (therapeutic vaccine) on livers of mice carrying HBV, which led to the observation of an induction in HBx-specific T-cell responses in combination with a significant depletion of HBsAg and HBV genome levels [219]. This is seemingly advantageous over previous studies and clinical trials of therapeutic vaccines against other HBV antigens, which showed limited success. Further, an application of these on already established liver failure might also result in a contrary effect [220,221].

### 5.4. HBx as Potential Therapeutic Target for Small Molecule Inhibitors

Alternatively, small molecules as antiviral agents become of great interest also in the field of chronic HBV treatment. In this regard, several compounds primarily acting by modulation of viral capsid assembling were already established and are partially under clinical trial with promising results for long-term treatment of chronically infected patients [222]. However, the development of chemical molecules targeting HBx as a therapeutic target still remain a challenge due to the lack of structural insights into HBx. Nevertheless, some studies used the HiBit-tagged HBx system for screening and identifying appropriate candidates. Hereby, Nitazoxanide was identified to efficiently inhibit the interaction of HBx with DNA binding protein 1 (DDB1). This compound hinders HBx–DDB1 interaction and restores the function of the SMC5/6 complex. In other studies, an inhibitor of NQO1 (Dicoumarol) prevents the recruitment of HBx to the cccDNA and therefore transcription of viral proteins [125,223]. In addition, epigenetic modifications by HBx are promising as a potential therapeutic target, but still need to be further investigated [224].

Overall, the therapeutic targeting of HBx alone or in combination with other established clinical applications could contribute to a functional cure of HBV-induced HCC, especially in the case of genomic integration and elevated HBx expression even prior to severe liver damage.

## 6. Final Remarks and Further Prospects

In summary, the identification of the hNTCP as primary entry receptor for HBV infections enabled the possibility for novel cell culture systems and especially infection models for HBV research. This breakthrough offered a tremendous opportunity over the past decade to improve understanding of the viral life cycle of HBV and includes particularly the investigation of HBx function. Difficulties in detecting HBx, low expression levels and solubility issues kept HBx controversially discussed for a long time, especially with respect to its function during HBV progression and contribution in liver disease progression. The pleiotropic behavior and interplay with multiple cellular proteins in the nucleus, the cytoplasm and by interaction with mitochondria demonstrate the crucial importance of HBx during HBV infection. In addition, the tremendous participation of HBx in initiation and progression of cellular transformation up to hepatocellular carcinoma development is still not fully understood and we witness the gap in current knowledge and pharmacologic prospective. However, the lack of a high-resolution structure poses a challenge to overcome this claim.

In addition, seroconversion is associated with the integration process and HBx expression, both hallmarks for tumor modulation and persistence of the infection. However, little is known about the contribution of HBx during different phases of infection, even though it is prospected as a better potential therapeutic target as compared to previous approaches. Importantly, the different HBV genotypes highly influence the course of disease and clinical outcomes in different ways. In light of this, HBx variants of different genotypes might also have a significant impact on the protein’s function and could have an important contribution in HBx-mediated host signaling pathways as well as the HBV life cycle. As this is not yet elucidated, but of fundamental interest to better understand the HBx-dependent role in hepatocarcinogenesis, research on this field is strongly needed.

## Figures and Tables

**Figure 1 ijms-24-04964-f001:**
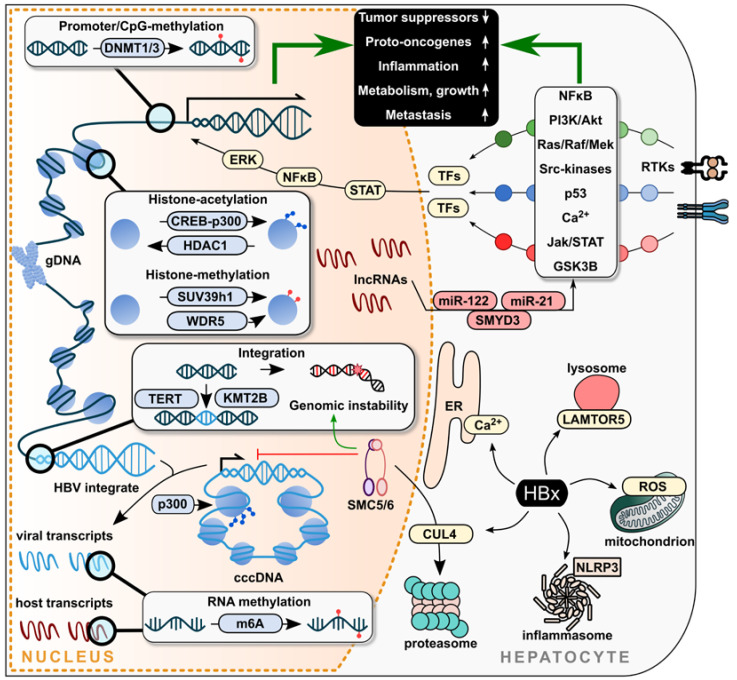
Schematic overview of HBx-mediated signaling pathways in the context of hepatocellular carcinoma progression. The multifunctional protein HBx deregulates several cellular pathways in the cytosol and nucleus, which are key factors participating in the development of HBV-induced liver pathogenesis and tumorigenesis. Nuclear pathways include the epigenetic modification by DNA methylation via DNMT1/3 as well as histone modifications via CREB-p300, HDAC1, WDR5 or SUV39h1. On the other hand, host and viral RNA methylation via m6A is also regulated by HBx. Additionally, HBx drives the expression of non-coding RNA (lncRNA and miRNA) such as miR-122, miR-21 or SMYD3. The integration of viral DNA into the host genome in proximity of the genes TERT, KMT2B or others enhances the pool of HBx transcripts and leads to genomic instability within the host genome. Cytosolic HBx mediates dysregulation of signaling pathways in hepatocytes, which is mediated by interaction with several cellular proteins of the ER, affecting cellular Ca^2+^ homeostasis at the lysosome (LAMTOR5) and mitochondria, thereby inducing ROS, the NLRP3 inflammasome and proteasomal adaptors such as CUL4. Thereby, it elicits expression of genes being regulated by, e.g., ERK, NF-κB, STAT or other TFs. Further, the interaction of HBx with SMC5/6 facilitates viral replication and plays a key role in DNA damage repair mechanisms. Ca^2+^, calcium ions; cccDNA, covalently closed circular DNA; CpG, cytosine-phosphate-guanosine; CREB, cAMP response element-binding protein; CUL4, cullin 4; DNMT1/3, DNA methyltransferases; ER, endoplasmic reticulum; ERK, extracellular signal-regulated kinase; gDNA, genomic DNA; GSK3B, glycogen synthase kinase-3 beta; HBV, hepatitis B virus; HDAC1, histone deacetylase 1; JAK, Janus kinase; KMT2B, lysine methyltransferase 2B; LAMTOR5, late endosomal/lysosomal adaptor, MAPK and mTOR activator 5; m6A, N6-methyladenosine methyltransferase; MAPK and MTOR activator 5; lncRNA, long non-coding RNAs; miRNA, micro RNA; NF-κB, nuclear factor kappa B; NLRP3, NLR family pyrin domain containing 3; p300, histone acetyltransferase p300; p53, tumor protein p53; PI3K, phosphoinositide-3-kinase; AKT, protein kinase B; Ras, rat sarcoma protein; Raf, rapidly accelerated fibrosarcoma protein; Mek, mitogen-activated protein kinase; ROS, reactive oxygen species; RTKs, receptor tyrosine-kinases; SMC5/6, structural maintenance of chromosomes 5/6; SMYD3, SET and MYND domain containing 3; Src, SRC proto-oncogene; STAT, signal transducer and activator of transcription 3; SUV39h1, SUV39H1 histone lysine methyltransferase; TERT, telomerase reverse transcriptase; TF, transcription factor; WDR5, WD repeat domain 5.

**Figure 2 ijms-24-04964-f002:**
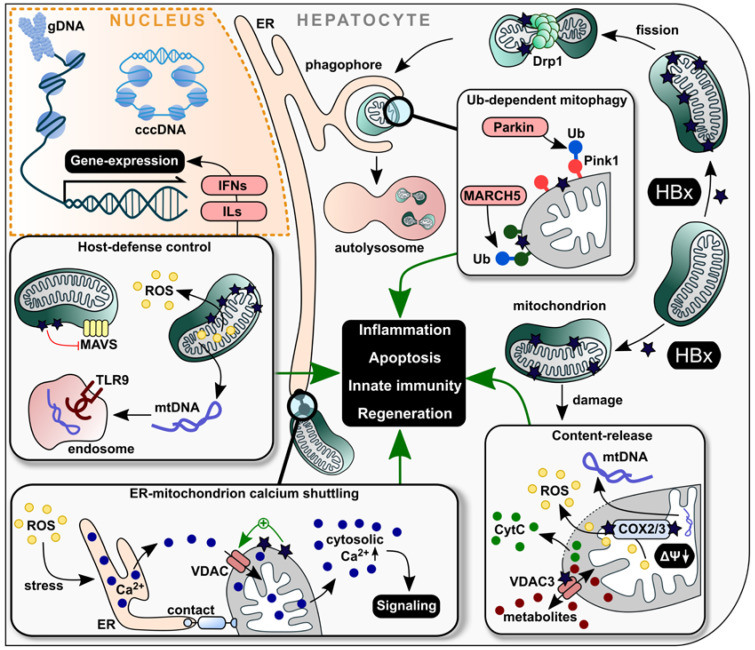
Schematic overview of HBx interaction with mitochondrial proteins in the context of liver pathology. HBx exerts fundamental effects on mitochondria themselves as well as on cellular processes linked to these organelles. Here, it modulates the mitochondrial morphology through Drp1 interaction towards fission, which leads to Ub-dependent mitophagy. At the OMM, interaction with both the ion channel VDAC3 and the redox chain proteins COX2/3 cause shifts in the mitochondrial membrane potential, followed by a release of CytC, ROS and mtDNA. This leads to an induction of apoptosis and inflammation. Furthermore, increased ROS levels induce ER stress and transport of Ca^2+^ through the ER, the mitochondrion and, finally, into the cytosol, which elevates associated signaling pathways. On the side of host defense modulation, the interaction of HBx with MAVS downregulates the antiviral immune response and upregulates pro-inflammatory mediators such as IFNs and ILs. Further, the release of damaged mtDNA contributes to the activation of several signaling pathways and, in particular, to an activation of the TLR9 receptor. Ca^2+^, calcium ions; cccDNA, covalently closed circular DNA; COX2/3, cyclooxygenase 2/3; CytC, cytochrome complex C; Drp1, dynamin-related protein 1; ER, endoplasmic reticulum; gDNA, genomic DNA; IFN, interferon; IL, interleukin, MARCH5, membrane-associated RING-CH 5 protein, MAVS, mitochondrial antiviral signaling protein; mtDNA, mitochondrial DNA; PINK1, PTEN-induced kinase 1; ROS, reactive oxygen species, TLR9, toll-like receptor 9; Ub, ubiquitin; VDAC3, voltage-dependent anion channel.

## Data Availability

No new data were created or analyzed in this study. Data sharing is not applicable to this article.

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
