# Peer review of "Relevance of HBx for Hepatitis B Virus-Associated Pathogenesis"

_ijms, 2023, doi:10.3390/ijms24054964_

Round 1

Reviewer 1 Report

Schollmeier et al., review the role of HBx in the pathogenesis of HBV infection.  The review is fairly extensive.  Comments and suggestions are appended below:

Major comments

1) A few lines on the role of HBx in regulating cell cycle could be added

2) The role of m6A in regulating HBx levels (PMID: 34851655) and the role of HBx in recruiting m6A writers to modify virus and host RNA (PMID: 33397803) are important developments in the field that may be discussed in the review. Perhaps the authors can consider including a figure on these recent findings.

3) Overall, the introduction to HBV basics appears to be very extensive and can be edited for brevity.

Minor comments:

1) “As of the latest GBD (Global Burden of Disease) report in 2019…” consider rephrasing

2) “All viral species comprise a genome being ~3.2 kilobases (kb) …” use “Members of the Orthohepadnavirus comprise…..”

3) “HBsAg (small surface antigen/protein) and (vii) HBxAg (regulatory X-protein). While  HBcAg can be N-terminally elongated to form HBeAg from one mRNA, S-HBsAg can be N-terminally elongated via the preS2 or preS1/S2 domains to form M-HBsAg or L-HBsAg  from another mRNA, respectively….”  - the word elongated does not sound appropriate and can be misleading.

4) “The remaining third displays classical signs….” – please use “display”

5) “Apart from the acute course of infection, HBV-infections carry a high risk of chronification,……”

6) “While HbsAg represents one of the major host-modulatory viral proteins,…” – please use “HBsAg” instead

7) “A strong bias on males being at risk of developing an HCC….” Please replace “on” with “of”

8) “Here, gtC and gtD are described to have the highest rate of promoting an HCC….”  - the word “an” should be deleted

9) “With respect to viral nucleic acids, a high genomic copy number within the serum of patients indicates a more severe progression of disease” the word “ within” should be replaced with“in”

10) “Interestingly, recent studies indicated…” – using “indicate” may be more appropriate

11) “HBV adds a significant pressure on genomic instability” – may be rephrased as “HBV significantly contributes to genomic instability”

12) “…….progression towards an HCC” – the word “an” should be deleted.

13) “Insertions are mainly sources of HBsAg, especially in case of HBeAg-negativity [47], and HBx….” The part of HBx is incomplete. This must be rephrased.

14) “Reason for this is a novel study suggesting that integration is more likely to be a side-effect of prolonged chronicity and not the driver of oncogenesis due to insertion at specific sites [51]” – this sentence needed to be rephrased.

15) “Just like HBeAg, the level of HBsAg in chronic HBV may be used as a certain type of prognostic marker” – please rephrase “certain type of prognostic marker”

 These minor edits are from the first 5-6 pages of the manuscript and the entire manuscript will benefit from English language editing. 

Author Response

Reviewer 1:

Major comments

Comment P1.1: A few lines on the role of HBx in regulating cell cycle could be added

Response: We agree with the reviewer on this important point and have incorporated this suggestion throughout our manuscript in line 546ff in the revised manuscript.

Comment P1.2: The role of m6A in regulating HBx levels (PMID: 34851655) and the role of HBx in recruiting m6A writers to modify virus and host RNA (PMID: 33397803) are important developments in the field that may be discussed in the review. Perhaps the authors can consider including a figure on these recent findings.

Response:  We thank the reviewer for providing this interesting query. We agree on the reviewer’s suggestion that m6A play and important role in the field of HBx mediated RNA modifications and have incorporated this aspect in line 458 ff in the revised manuscript. However, we believe that an additional figure about the role of m6A modifications on viral and host RNA would be outside of the scope of our review article, because the focus of our paper is set on the overall roles of HBx in the context of HCC progression and liver pathogenesis without a specific focus on single interaction modes e.g. m6A modifications. However, we have incorporated an aspect of this in the revised figure 1 and the respective figure legend. We hope that this presents as an acceptable compromise to the reviewer.

Comment P1.3: Overall, the introduction to HBV basics appears to be very extensive and can be edited for brevity.

Response: We agree on this point and deleted some sections in the introductory paragraphs. We hope that the extent of deletions is satisfactory, as we had the feeling that some aspects needed to be mentioned in order to set the relevance of HBx into a fitting context.

Minor comments:

Comment P1.1: “As of the latest GBD (Global Burden of Disease) report in 2019…” consider rephrasing

Response: OK.

Comment P1.2: “All viral species comprise a genome being ~3.2 kilobases (kb) …” use “Members of the Orthohepadnavirus comprise…..”

Response: The suggested “members of the Orthohepadnavirus” have exchanged the part “all viral species”.

Comment P1.3: “HBsAg (small surface antigen/protein) and (vii) HBxAg (regulatory X-protein). While  HBcAg can be N-terminally elongated to form HBeAg from one mRNA, S-HBsAg can be N-terminally elongated via the preS2 or preS1/S2 domains to form M-HBsAg or L-HBsAg  from another mRNA, respectively….”  - the word elongated does not sound appropriate and can be misleading.

Response: We have removed the word “elongated” in line 75ff and inserted a more precise formulation in this context.

Comment P1.4: “The remaining third displays classical signs….” – please use “display”

Response: OK.

Comment P1.5: “Apart from the acute course of infection, HBV-infections carry a high risk of chronification,……”

Response: We are uncertain about the suggestion for change, yet thought that the sentence may have been a bit unclear in its second part. We therefore changed it to “…, especially if infection occurs at a very young age” to emphasize the situation in HBV-infected infants in line 100 in the revised manuscript.

Comment P1.6: “While HbsAg represents one of the major host-modulatory viral proteins,…” – please use “HBsAg” instead

Response: OK.

Comment P1.7: “A strong bias on males being at risk of developing an HCC….” Please replace “on” with “of”

Response: OK.

Comment P1.8: “Here, gtC and gtD are described to have the highest rate of promoting an HCC….”  - the word “an” should be deleted

Response: OK.

Comment P1.9: “With respect to viral nucleic acids, a high genomic copy number within the serum of patients indicates a more severe progression of disease” the word “ within” should be replaced with“in”

Response: OK.

Comment P1.10: “Interestingly, recent studies indicated…” – using “indicate” may be more appropriate

Response: We agree and have incorporated your suggestion accordingly.  

Comment P1.11: “HBV adds a significant pressure on genomic instability” – may be rephrased as “HBV significantly contributes to genomic instability”

Response: The sentence was rephrased accordingly.

Comment P1.12: “…….progression towards an HCC” – the word “an” should be deleted.

Response: OK.

Comment P1.13: “Insertions are mainly sources of HBsAg, [47], and HBx….” The part of HBx is incomplete. This must be rephrased.

Response: We have removed the term “especially in case of HBeAg-negativity” to establish a clearer focus of the statement.

Comment P1.14: “Reason for this is a novel study suggesting that integration is more likely to be a side-effect of prolonged chronicity and not the driver of oncogenesis due to insertion at specific sites [51]” – this sentence needed to be rephrased.

Response: We rephrased the sentence in line 206ff in the revised manuscript and hope that this clarifies the point.

Comment P1.15: “Just like HBeAg, the level of HBsAg in chronic HBV may be used as a certain type of prognostic marker” – please rephrase “certain type of prognostic marker”

Response: Corrected. We have rephrased the term “a certain type” in line 224ff of the revised manuscript and used now the term “potential diagnostic and prognostic marker”.  

Reviewer 2 Report

The authors review different aspects of HBV infection with a particular emphasis on HBx protein, its role in viral replication, regulation of transcription, and adverse effects on cell physiology which result in development of liver cancer and liver damage.

The review is very large and extensive. Many aspects of HBV infection are very professionally summarized, with new, recent results highlighted. The Figures are new and provide the whole picture of HBx-related effects. 

The apparent drawbacks are that it is not very focused, and the main idea is floating around, but is not concentrated. This is likely not reparable, but I would still suggest to remove unnecessary parts and too much details on something not related to HBx.

Other major issues:

(1) Chapter 5 is chaotic and requires re-writing;

(2) Many typos and mispellings, English needs extensive editing

(3) Figure 1 requires addition of HBx-related m6A post-transcriptomic effects

Some of other issues in an orderly manner are outlined below: 

Line 6: please, remove “still”. It is a major global health problem

Line 8: an extra space should be removed before “caused”

Lines 27-28: this statistic is new to the Reviewer. The prevalence is typically estimated at 250 million cases with over 1 million deaths per year. The reference is missing as well

Lines 28-29: this statement should be referenced. It is particularly important because in many countries the incidence of acute hepatitis B cases were reduced, but chronic hepatitis B cases remained at the same level.

Lines 31-32: it appears not quite relevant to compare geographical regions with “income”. Please, rephrase.

Line 33: first should come the term, then its abbreviation. That is, years of life lost (YLL)

Line 35: it probably should be phrased as “hepatitis B VIRUS-related health complications”

Line 36: cirrhosis and liver disease – this phrase is not valid. Liver disease is HepB per se

Line 53: NTCP receptor. And again, first the term, next its abbreviation

Line 55: rcDNA, the same critique

Line 60: cccDNA, the same critique

Line 64: the Reviewer is not aware of cccDNA formation from dslDNA

Line 179-181. This sentence needs rephrasing. It is not harder to detect, but it is a harmful and unnecessary procedure to make a liver biopsy for detecting HBV cccDNA.

Line 261: HBx gene was suspected to encodes

Lines 276-277: this is a small protein with dynamically changing structure. Please, find appropriate literature and terms for this statemen

Line 283: an extra space

Line 304: an extra space

Line 321-323: there are infection models: HepG2-hNTCP, HepaRG, HepaRG-NTCP, PHH (these and other models are further described by the authors). As such, this statement is incorrect

Line 336: what is the purpose of mentioning mouse AML12 here? It is murine cell line

Line 337: the authors do not specify the BEL7404 cell line characteristics. What is it used for? It is not a common model

Line 342: an extra space

Line 345: there are also pigs, dogs etc which overexpress human hNTCP (please, see one of the publications by S. Urban)

Line 345: as the authors may know, research in chimpanzees is banned by NIH, so they should remove them from here or mention this fact

Line 357: an extra space

Line 357: the term for this is 1.5mer

Liness 357-358: it is unclear what “cell line-dependent diversity of HBx” stands for

Lines 360-361 are unclear. Why they mention anti-HBx antibodies?

Line 365: an extra space

Line 368: an extra space

HBx also affects cccDNA co-transcriptionally by installing m6A post-transciptomic marks, regulating pgRNA and HBV RNAs stability, as well as reverse transcription to rcDNA. These should be added. Along with that, there are reports on HBx-m6A mediated HCC development. These studies should be mentioned.

HBx effects on HBV RNA post-transcriptional states and m6A-driven HCC development should be added to Figure 1

Line 603: what is WLR68 cell line? What is the rationale of mentioning it here?

The authors use a lot of abbreviations, many of which are used only once. Please, check and avoid any unnecessary terms and abbreviations

Line 751: what means “blocking SMC5/6 complex”? It is destroyed by HBx; the therapeutic rationale is to restore SMC5/6 which may interact with cccDNA and silence it transcriptionally

Lines 779 -781: this activity is related to the restoration of SMC5/6. The small molecule was designed to inhibit HBx-DDB1 interaction. The authors should correct these sentences and make this clear

The whole chapter 5 seems fairly chaotic, with many distantly related parts being merged together, and many close studies being separated by several paragraphs. Please, modify this for clarity, combine related approaches, and make clear statements regarding the prospects of these methods.

Author Response

Reviewer 2:

Comment P2.1: Chapter 5 is chaotic and requires re-writing;

Response: We do see the issue of chapter 5 appearing slightly hard to read. However, we tried to summarize overall aspects of the manuscript in this section and wanted to focus the reader on open questions and novel findings. In order to give this section a clearer order, we inserted subheadings to each section dealing with a different aspect in the revised manuscript. We hope that this presents as satisfactory solution.

Comment P2.2: Many typos and mispellings, English needs extensive editing

Response: Thank you for your comments on the language. We have had edited this article by aa native  speaker to correct all typos and misspellings accordingly.

Comment P2.3: Figure 1 requires addition of HBx-related m6A post-transcriptomic effects

Response: Along with suggestions made by reviewer 1, we have incorporated this aspect, along with viral RNA methylation, in the revised figure 1 and the respective figure legend in the updated manuscript (line 594ff).

Some of other issues in an orderly manner are outlined below:

Comment P2.1: Line 6: please, remove “still”. It is a major global health problem

Response: OK.

Comment P2.2: Line 8: an extra space should be removed before “caused”

Response: OK.

Comment P2.3: Lines 27-28: this statistic is new to the Reviewer. The prevalence is typically estimated at 250 million cases with over 1 million deaths per year. The reference is missing as well

Response: These information’s present the latest global data and are based on the aforementioned GBD report, which is cited twice in this paragraph. To clarify that all of the informations given in this paragraph are based on the GBD report from 2019, notes have been added in line 30 and line 36 of the revised manuscript.

Comment P2.4: Lines 28-29: this statement should be referenced. It is particularly important because in many countries the incidence of acute hepatitis B cases were reduced, but chronic hepatitis B cases remained at the same level.

Response: These data are based on the GBD report from 2019, which are referenced. Changes were made according to the reply to comment P2.3.

Comment P2.5: Lines 31-32: it appears not quite relevant to compare geographical regions with “income”. Please, rephrase.

Response: This comparison is based on the GBD report from 2019. It may appear distorting to the readers if changes to the classification made in this report were to be applied, which is why we refrain from incorporating any changes. We hope that the reviewer accepts this.

Comment P2.6: Line 33: first should come the term, then its abbreviation. That is, years of life lost (YLL)

Response: We agree and apologize for this formal issue. We have corrected the abbreviation in line 37 accordingly and also proofed and adjusted the following abbreviations.

Comment P2.7: Line 35: it probably should be phrased as “hepatitis B VIRUS-related health complications”

Response: The word “virus” has been included in the revised manuscript.

Comment P2.8: Line 36: cirrhosis and liver disease – this phrase is not valid. Liver disease is HepB per se

Response: We agree that the term “liver disease” is misleading in this context and have removed the part in the revised manuscript.

Comment P2.9: Line 53: NTCP receptor. And again, first the term, next its abbreviation

Response: OK.

Comment P2.10: Line 55: rcDNA, the same critique

Response: OK.

Comment P2.11: Line 60: cccDNA, the same critique

Response: OK.

Comment P2.12: Line 64: the Reviewer is not aware of cccDNA formation from dslDNA

Response: This is referred to as illegitimate replication and is mediated via non-homologous recombination (Yang et al., 1995 and 1998), yet we agree that this is not majorly contributing to the formation of functional cccDNA. Thus, we adjusted the statement in line 67ff in the revised manuscript.

Comment P2.13: Line 179-181. This sentence needs rephrasing. It is not harder to detect, but it is a harmful and unnecessary procedure to make a liver biopsy for detecting HBV cccDNA.

Response: We agree with this point and corrected the term accordingly in the revised manuscript.

Comment P2.14: Line 261: HBx gene was suspected to encodes

Response: The “s” in “encodes” was removed.

Comment P2.15: Lines 276-277: this is a small protein with dynamically changing structure. Please, find appropriate literature and terms for this statemen

Response: We are sorry for this misunderstanding. We did not want to imply the separation into classical subdomains, yet the separation into certain parts such as C- or N-terminal regions. Functions of distinct parts of HBx are set forth in the following section(s) with plenty of references. The sentence was modified in line 289f in the revised manuscript.

Comment P2.16: Line 283: an extra space

Response: OK.

Comment P2.17: Line 304: an extra space

Response: OK.

Comment P2.18: Line 321-323: there are infection models: HepG2-hNTCP, HepaRG, HepaRG-NTCP, PHH (these and other models are further described by the authors). As such, this statement is incorrect

Response: We agree that there are infection models to investigate HBV specific characteristics, however in this context, due to the prefixed word “long-standing” we wanted to highlight the situation in HBV research before the description of NTCP as viral entry receptor and other mentioned cell lines where investigated as suitable application for HBV infection. We have changed the word “lack” with “limitations in infections models” in line 335f of the revised manuscript to clarify this statement and hope that the edited section clarifies this point.

Comment P2.19: Line 336: what is the purpose of mentioning mouse AML12 here? It is murine cell line

Response: We agree that a mouse cell line is not the most common model for hNTCP overexpression to gain HBV susceptibility. We have deleted this cell line from the sentence in the revised manuscript.

Comment P2.20: Line 337: the authors do not specify the BEL7404 cell line characteristics. What is it used for? It is not a common model

Response: The reviewer raised an important point, that the mentioned cell line is not a common model for HBV research. However, the scope of this statement is also the reflection of novel culture models for HBV research besides the common models. The referred paper about the usage of BEL7404 cells was published in 2022 and provides a novel approach of a new cell culture system for HBV replication and infection because of several advantages of the used cell line. According to the publication, the cell line demonstrates higher levels of HBV antigen expression after transfection. Based on this, the expression of HNF4α was reported to promote pgRNA synthesis and offers, together with NTCP expression, a robust system to study HBV infection and replication. To clarify this aspect, we have redrafted the section in line 352ff of the revised manuscript to highlight the advantage of the mentioned cell line.

Comment P2.21: Line 342: an extra space

Response: OK.

Comment P2.22: Line 345: there are also pigs, dogs etc which overexpress human hNTCP (please, see one of the publications by S. Urban)

Response: A note along with an additional reference has been added to this sentence in line 364 in the revised manuscript.

Comment P2.23: Line 345: as the authors may know, research in chimpanzees is banned by NIH, so they should remove them from here or mention this fact

Response: According to the reviewer’s suggestion, we have been removed the “chimpanzees” from the list of appropriate animal models to study HBV infection since the research is banned by the NIH.

Comment P2.24: Line 357: an extra space

Response: OK.

Comment P2.25: Line 357: the term for this is 1.5mer

Response: OK.

Comment P2.26: Liness 357-358: it is unclear what “cell line-dependent diversity of HBx” stands for

Response: We have to clarify that “cell line dependent diversity” means that the investigations of HBx in a transfection system strongly depends on the cell line used for expression. We have redrafted the section by using the term “cell line specific effects” and hope, that the correction clarifies the point we attempted to make.

Comment P2.27: Lines 360-361 are unclear. Why they mention anti-HBx antibodies?

Response: This paragraph deals with obstacles in studying the functions of HBx. No proper antibodies against the protein are available on the market, which certainly presents as obstacle. An additional statement was added to the sentence in line 379ff of the revised manuscript.

Comment P2.28: Line 365: an extra space

Response: Unfortunately, we could not identify an extra space in this line.

Comment P2.29: Line 368: an extra space

Response: OK.

Comment P2.30: HBx also affects cccDNA co-transcriptionally by installing m6A post-transciptomic marks, regulating pgRNA and HBV RNAs stability, as well as reverse transcription to rcDNA. These should be added. Along with that, there are reports on HBx-m6A mediated HCC development. These studies should be mentioned.

Response: Please see major comment P1.2.

Comment P2.31: HBx effects on HBV RNA post-transcriptional states and m6A-driven HCC development should be added to Figure 1

Response: Please see major comment P2.3.

Comment P2.32: Line 603: what is WLR68 cell line? What is the rationale of mentioning it here?

Response: These were used in the cited publication, which is why they are mentioned there.

Comment P2.33: The authors use a lot of abbreviations, many of which are used only once. Please, check and avoid any unnecessary terms and abbreviations

Response: We partially agree on this point, yet we feel that not every reader is familiar with the full name of certain genes and vice versa. For the sake of completeness, we would like to keep the abbreviations as they are and hope the reviewer understands our concern.

Comment P2.34: Line 751: what means “blocking SMC5/6 complex”? It is destroyed by HBx; the therapeutic rationale is to restore SMC5/6 which may interact with cccDNA and silence it transcriptionally

Response: We agree with the reviewer about the confusing wording and have replaced the word “blocking” by “restoring” in the revised manuscript and add a sentence for clarification of this statement in line 790f.

Comment P2.35: Lines 779 -781: this activity is related to the restoration of SMC5/6. The small molecule was designed to inhibit HBx-DDB1 interaction. The authors should correct these sentences and make this clear

Response: We agree on additionally mentioning SMC5/6 specifically in line 818f of the revised manuscript.

Comment P2.36: The whole chapter 5 seems fairly chaotic, with many distantly related parts being merged together, and many close studies being separated by several paragraphs. Please, modify this for clarity, combine related approaches, and make clear statements regarding the prospects of these methods.

Response: Please see major comment P2.1

Round 2

Reviewer 2 Report

The authors corrected the issues and improved the manuscript accordingly. I recommend it for publication